# Sink Strength Maintenance Underlies Drought Tolerance in Common Bean

**DOI:** 10.3390/plants10030489

**Published:** 2021-03-05

**Authors:** Amber Hageman, Elizabeth Van Volkenburgh

**Affiliations:** Department of Biology, University of Washington, Seattle, WA 98195, USA; ahageman@uw.edu

**Keywords:** SUT, proton pump, yield, pod harvest index, harvest index, partitioning, allocation

## Abstract

Drought is a major limiter of yield in common bean, decreasing food security for those who rely on it as an important source of protein. While drought can have large impacts on yield by reducing photosynthesis and therefore resources availability, source strength is not a reliable indicator of yield. One reason resource availability does not always translate to yield in common bean is because of a trait inherited from wild ancestors. Wild common bean halts growth and seed filling under drought and awaits better conditions to resume its developmental program. This trait has been carried into domesticated lines, where it can result in strong losses of yield in plants already producing pods and seeds, especially since many domesticated lines were bred to have a determinate growth habit. This limits the plants ability to produce another flush of flowers, even if the first set is aborted. However, some bred lines are able to maintain higher yields under drought through maintaining growth and seed filling rates even under water limitations, unlike their wild predecessors. We believe that maintenance of sink strength underlies this ability, since plants which fill seeds under drought maintain growth of sinks generally, and growth of sinks correlates strongly with yield. Sink strength is determined by a tissue’s ability to acquire resources, which in turn relies on resource uptake and metabolism in that tissue. Lines which achieve higher yields maintain higher resource uptake rates into seeds and overall higher partitioning efficiencies of total biomass to yield. Drought limits metabolism and resource uptake through the signaling molecule abscisic acid (ABA) and its downstream affects. Perhaps lines which maintain higher sink strength and therefore higher yields do so through decreased sensitivity to or production of ABA.

## 1. Introduction

Common bean (*Phaseolus vulgaris* L.) is grown around the world as a staple crop and is an important source of dietary protein for more than 300 million people [1]. However, each year, more than 60% of bean crop production is affected by water deficits [2]. Those reductions of yields range from minor to complete loss. Much of common bean production is on small-holder farms in areas where droughts are common and access to irrigation is limited [3]. This can have large impacts on food security in these regions. Droughts are only expected to increase in much of these growing areas [4], underlining the importance of better understanding drought tolerance in common bean.

The severity of yield losses depends on many factors, including genotype, developmental timepoint at onset of drought [5], soil type and other climatic conditions [6]. Drought has an impact on plants by limiting photosynthesis and therefore depriving the plant of building blocks necessary for growth and reproduction. However, correlations between photosynthesis and yield are not strong [7] particularly under suboptimal conditions [8], and plants may fail to set and fill seeds, even when not limited by photosynthetic resources [9]. While this phenomenon is much less understood, one important reason it may occur in common bean is the particular natural history and domestication path of bean.

As humans bred wild bean plants into domesticated forms, we altered both their genetics and growth environment in ways which made their native physiological responses to stress less important and even detrimental to yield [6]. For example, common beans toggle between a faster growth phase when conditions are favorable and a slower phase when they are not. While this trait is beneficial for wild, indeterminate bean plants, the persistence of this trait in domesticated species can prevent us from maximizing crop yields. The reason for this is that domestication of common bean has selected for determinate growth [10]. This bush growth habit resulted in more synchronized flowering and fruit set which uses up all reproductive potential at once. However, domestication did not eliminated the tendency for determinate bean plants which are producing reproductive structures to toggle back to slow growth during stress conditions [4]. As a result, allocation of resources to developing flowers, pods, and seeds may be interrupted during a drought episode leading to flower and pod abortion, or failure to fill seeds fully. Coupled with the inability to produce new flowers that came with determinacy, this tendency makes it so their one opportunity at reproduction may be largely diminished [4].

With the goal of increasing yield, researchers discovered that leaf area index, which quantifies leaf area per unit of land, and canopy biomass accumulation stand out as a strong indicators of yield [11,12,13,14], even above net assimilation rates [15]. Similarly, direct relationships between yield and rates of leaf production and expansion have also been seen in crops [16] and leaf size correlates with seed size in common bean [17]. Further, studies show harvest indexes (HI), which measure the efficiency in which a plant turns biomass into seeds, are an even stronger predictor of yield [2,12]. Increased HI has been increasingly selected for in breeding of crops, including common bean, for drought tolerance [18]. In common bean, a trait that quantifies how much of the total pod biomass is partitioned to seeds at maturity, called Pod Harvest Index (PHI), is consistently the strongest predictor of yield under both well-watered and droughted conditions in common bean [11]. Therefore, genotypes which continue filling seed under drought, rather than toggling back into to slower growth as wild beans do, are the lines that fare best under drought. Together, these findings show a tight linkage between growth, allocation and yield.

Focusing in on these findings, a new study measured and tested correlations between leaf growth rates and PHI across 20 common bean genotypes [19]. It showed that leaf growth rates and harvest indices are both similarly affected by drought within a genotype—that is, genotypes that slow their leaf growth strongly under water deficit also slow their allocation of biomass towards yield. Conversely, genotypes which maintained leaf growth under drought also maintained allocation towards seeds. That these two processes, leaf growth and biomass allocation to yield, are similarly affected by drought, and that both are strong predictors of yield under drought, led us to ask what common mechanisms control growth and allocation processes and how does drought affect those commonalities. Understanding how these responses differ within drought-tolerant and drought-sensitive lines could allow breeders to find novel solutions to help beans overcome their native drought response.

## 2. Limitations on Growth

The ability to continue growing leaves and seeds in common bean and other crops could be hindered by one of two big picture limitations, or both: source limitations or sink limitations [20,21]. Sources are any tissue that provides resources for export to the rest of the plant—photosynthetically active leaves, as well as storage tissues such as stems or roots. Sinks on the other hand are tissues that take in resources for growth, maintenance or storage, such as immature leaves, flowers and seeds. Resources flow between sources and sinks through the phloem according to concentration gradients (traveling from high to low) and resistances (such as membranes). The ability for sources to export and sinks to competitively import resources is quantified as source and sink strength, respectively [20]. Growth rates of sinks are a good reflection of sink strength [22], even though some resources are used up in respiration and therefore do not add to dry weight gain [20].

Sources can limit growth by not having enough resources to export, or by not exporting them even if they are available. The best example of source limitation is caused by the inhibition of photosynthesis. One reasons this can occur is when levels of abscisic acid (ABA), a well-studied stress-related phytohormone [23], increase under drought. ABA acts as a stress signal, increasing under a number of stressors but most strongly under drought, and helps plants sense their environment and adjust their physiology accordingly [24]. The mechanism of ABA action includes inhibition of the plasma membrane proton pump, leading to loss of cellular K^+^ to the apoplast, resulting in a decrease of turgor of guard cells and stomatal closure. Stomatal resistance slows the rate of photosynthesis by reducing carbon dioxide diffusion to the chloroplasts. This in turn reduces the amount of carbohydrate fixed in the source tissues, and thus source strength [25]. Source strength may also decrease under drought, since enzymes which are involved in sucrose synthesis are inhibited which can lead to less sucrose available for export [26].

However, even though sources can limit sink development, there are many examples that show sink growth and development are not directly or exclusively linked to source strength. One simple demonstration of this independence is that once seed filling has begun, dry matter accumulation remains linear even when source:sink ratios change [8]. Many crops are able to remobilize stored resources under drought to compensate for reduced photosynthetic fixation [27]. Instead, a decrease in available resources will decrease seed filling duration rather than rate [28]. Other examples show that in common bean, apoplastic sucrose levels surrounding the seed do not correspond to sucrose uptake rates within those seeds [29]. Therefore, even though seeds may have access to resources, they do not necessarily take them up. Likewise, in grain legumes, defoliation rates (33%, 66% and 99%) do not proportionally predict yield losses (20%, 32% and 35%, respectively) [30]. Additionally, “stay-green” genotypes, whose leaves senesce later than other genotypes and do more photosynthesis as a result, do not necessarily have increased yields—indeed, yields are actually lower in many stay-green genotypes under drought when compared to non-stay-green genotype [31,32]. When considering whether leaf growth rates of immature sink leaves are also decoupled from source strength, evidence suggests sink leaves also exert control over their growth independent from source availability. Leaf disk studies in common bean indicate that light, decoupled from photosynthesis, acts as a stronger stimulator of growth than does sucrose, pointing to regulatory mechanisms playing a larger role than simply substrate availability [33]. Studies also show that partial defoliation in common bean leads to increased photosynthetic rates of the remaining leaves in order to meet the demand of the sink leaves [34]. Since source strength was able to increase, this indicates sink limited growth in immature leaves [35]. Additionally, in another study, partial defoliation of common bean actually lead to increased growth rates of remaining leaves [36]. Lastly, in two common bean genotypes whose growth rates are high and low, comparatively, photosynthetic rates do not differ significantly on a per area basis (Banerjee and Van Volkenburgh, unpublished).

Therefore, it is clear that sinks exert fine control over their ability to acquire resources and grow when source limitations are not hindering sink growth, while sources exert a more course control [37]. Sinks exert this control by either changing uptake into the sink or metabolism within the sink; both result in reducing the gradient that acts as the driving force for continued resource delivery from sources. As mentioned above, in common bean lines which are filling seeds, two different lines can have the same supply of sugar available to the seeds yet different rates at which they take up the sugar [29]. Growth rates of those lines’ cotyledons, cultured in vitro, retain differences between genotypes, suggesting filial tissues themselves (the seeds) have control over filling rates (sink strength). These differences, which cannot be fully explained by differences in seed surface area, may be controlled genetically within the seed and play an important role in determining yield. This conclusion has been reached by others, who find that yield, seed growth rates, and general sink strength are typically sink limited [38,39]. What then controls sink strength and how is this affected by drought?

## 3. Control of Sink Strength

Since at least the late nineties [40], it has been speculated that the growth of a sink leaf may be most directly regulated by the ability of its individual cells to expand, rather than the carbohydrate supply available to the sink. This expansion is necessary for cell division in early development, setting up a strong gradient to continue drawing resources to that organ. Cell expansion in leaves is thought to be controlled by the acid-growth mechanism in which an acidic environment in the cell wall results in increased loosening of the wall via expansins, which in turn allows it to be extended by turgor pressure within the cell [41,42]. Cell wall acidification is set up via H^+^-ATPases actively pumping protons out of the cell and into the apoplastic space. The pH gradient resulting from acidification of the cell wall set up by the proton pumps also acts as the driving force bringing resources in (carbon, nitrogen) via secondary active transport. This then affects metabolism and turgor pressure through changes to solute and ion concentrations within the cell.

We suggest that all sink growth, whether that sink is a leaf, fruit or seed, is primarily controlled by cell expansion. Indeed, recent work continues to highlight the significance of cell wall loosening and expansion for regulating sink strength in tissues beyond leaves [43,44]. Acidification, which fuels loosening and expansion, also increases uptake rates of sucrose and nitrogen (and therefore metabolism) as well as other solutes (and therefore turgor). If all sinks grow via the same mechanism, it seems likely that all sink tissue growth within a single plant would relate to one another. This would explain the findings mentioned above that leaf size is positively correlated with seed size [45] and more generally, that different tissues’ growth, size and weight gain all positively correlate within a genotype [17]. Additionally, breeders consistently find that traits which measure carbohydrate allocation to seeds, harvest index and pod harvest index in bean, have the strongest correlation with yield across all conditions and genepools [2]. Therefore, when differences exist in yield between different genotypes that is not described by differences in photosynthate availability, this may be due to differences in sink strength between genotypes. Control could stem from differences in the ability to regulate (1) pH of the cell wall via proton pumping, (2) carbohydrate and ion uptake via transporters and channels, and/or (3) metabolism via enzymes. These three mechanisms all offer important points of regulation that could originate from within sink cells themselves to control leaf sink strength via cell expansion.

However, while all three of these mechanisms are vital for cell expansion, we believe resource uptake rates, particularly carbohydrate uptake rates, exert the largest control over a cell’s ability to grow. Although regulation of uptake, turgor and pH shows many interconnections, the simple result that differences in *Phaseolus vulgaris* seed growth rates were accounted for by dry matter weight gain and size [46], which in turn were found to be determined by sucrose uptake, suggest that regulation of uptake is at the core. In bean and many other crops, this is accomplished via sucrose uptake co-transporters, SUCs (sometimes called SUTs), which actively co-transport sucrose with protons into the cell [47]. Genotypes with the highest growth rates had highest maximum SUC rate, highest SUC protein levels, and the highest proton pump protein concentrations [29]. Carbon uptake itself exerts control over pH since sucrose is symported into the cell with protons. This means increased uptake of carbon also leads to increased wall pH. Expressing more proton pumps could counteract this affect and keep the wall acidic however. Additionally, both of these mechanisms affect turgor through changing osmolytes in the cell as well as cell wall loosening via pH and therefore expansion. How then is carbohydrate uptake regulated?

Carbon, which makes up 90% of both a plant’s overall mass and bean seed dry weight [48,49], reaches the sink apoplasm through the phloem. At the apoplast/cell interface, uptake of carbohydrates, often in the form of sucrose, must occur across the plasma membrane. SUCs rely on a proton gradient across the membrane to fuel this movement, set up by a proton pump on the seed membrane. Sucrose may also enter a sink cell via an unsaturable, passive mechanism however, probably the sucrose gradient is not sufficient to drive this transport mechanism. Instead, secondary active transport predominates in determining the total rate of uptake. Differences in sucrose uptake rates between genotypes likely arises from differences in densities of SUCs or proton pumps. Strong correlations (r^2^ = 0.95) between maximum SUC rates and sucrose uptake, normalized by size, support this hypothesis [29]. Further, sucrose uptake correlates strongly with SUT1 protein levels, suggesting that when the protein is present, it is actively working to take up sucrose [29].

Densities of SUCs and proton pumps are regulated within sink tissues themselves. Increasing sucrose import into seeds leads to an increased metabolism of both C and N within seeds, resulting in a positive feedback loop which maintains continued uptake of both. When sucrose concentrations decrease within the sink, sugar sensing mechanisms detect this drop and a signaling cascade results in increased SUC transcription [50] (Figure 1). With more SUCs comes an increase in C uptake into the seed, which lowers the sucrose concentration in the apoplast and increases water potential. This leads to a loss of water from the apoplast to surrounding tissues, the seed coat in bean, increasing turgor pressure within the seed coat. Increased turgor upregulates sucrose facilitators, which allow sucrose release from maternal tissues into apoplast [49,51]. Therefore, more C enters the apoplast and can be taken up into the seed, leading to the increases of C and N metabolism again. Increased phloem and seed loading of nitrogen in the form of amino acids results in increased SUC concentrations as well as root nitrate transporters [52], showing nitrogen and carbon metabolisms are highly linked. Additionally, import of carbohydrate is increased via upregulation of starch biosynthesis, which fits with the sugar sensing feedback loop described above [43]. Conversely, increased sucrose concentration correlates highly with decreases SUT1 expression [50].

## 4. Impacts of Drought on Sink Strength via ABA

Water deficits in the soil lead to decreases in water content within plants. This drop in water availability has the potential, depending on severity and developmental timepoint, to affect nearly every aspect of growth and development, including growth rates and cell turgor (loss of turgor leads to wilting). Interestingly, growth rates actually slow before loss of turgor in cells, showing that decreased growth is not a direct result of decreased turgor [53,54,55,56]. Additionally, drought does not result in evenly distributed water deficit within a plant. For example, after 6 days of water withholding, soy bean leaf water potential dropped steeply, however, seed water potential was buffered from the effects of water deficit, having not changed from well-watered conditions [57]. In addition, under drought conditions, older source tissues generally are the first to wilt and abscise. This allows resources to be remobilized out of these older tissues into newer growth, which either has more potential for future resource gathering or for reproductive output. Sink tissues are generally protected from the impacts of drought over other tissues [28], yet their growth still slows even when buffered against water and resource limitations. Therefore, with resource limitations and water availability not acting as primary regulators of sink growth under drought, what effects changes in physiology?

Physiological responses to drought have been largely attributed to the signaling hormone abscisic acid, ABA. Although ABA response is most classically associated with decreases in stomatal aperture, and therefore reduction to photosynthetic rates, ABA has also been shown to impact growth and development through other signaling pathways [57]. Of note, sometimes ABA stimulates growth, like it does in roots and in seedlings, while other times it inhibits growth, complicating how we understand both ABA action and the regulation of growth-related processes [58]. This phenomenon is not unique to ABA signaling, since drought generally has an opposite effect on root vs. shoot metabolic responses [59]. However, when we narrow our focus to above ground sinks, such as growing leaves or seeds, ABA usually acts as a growth inhibitor.

ABA inhibits growth by affecting multiple mechanisms related to resource uptake. In droughted wheat plants, ABA accumulates and correlates with decreases in seed set [60]. This may be due to ABA’s known impact on increasing cell-wall invertase (CWIN) inhibitors, since CWIN mutants have been shown to have small seeds [61] and seed size correlates with CWIN levels in seed coats [62]. CWIN hydrolysis of sucrose is essential for maintaining the sucrose gradient which drives normal carbon flux, therefore decreasing its activity would reduce carbon flow. ABA is also known to reduce cytokinins, phytohormones which promote growth through activating CWIN and cell cycle genes [63]. During the seed cell division and expansion phase, reduced cytokinins can result in full abortion of the seed [28]. ABA also regulates processes such as plant water balance and osmotic regulation in response to stress [64]. This may be through affects to water hydraulics via changes to water permeability through regulation of aquaporins [65].

ABA may also be an important regulator of resource uptake through its ability to inactivate some H^+^-ATPases [66]. In *Arabidopsis*, there are proton pumps unique only to the seed tissue, allowing for the possibility that control may stem from regulation of tissue specific proton pumps under drought [67]. These may be differentially impacted by ABA, changing proton motive forces and therefore carbon uptake into different tissues. Magnitudes of proton motive forces in general have been shown to estimate accumulation rates of sucrose in sinks [68], further supporting this theory. Proton pumps also differ in their activity depending on the developmental stage of the tissue. In mature leaves, activity is largely restricted to the guard cells and phloem [69], while activity is maintained throughout development in seed tissues [67].

Even in examples where ABA does not inhibit proton pumping, ABA plays an essential role in translating changes in pH into action. Drought causes apoplastic pH to increase and this in turn reduces wall extensibility and growth [55]. This can result in increased turgor, which helps cells maintain structure. Yet, in barley leaves, drought still induced apoplastic pH increases in ABA mutants. This shows ABA is not necessary to increase pH of the apoplast. However, leaf growth did not slow unless ABA was suppled [53]. This suggests ABA did not affect pH directly by inhibiting proton pumping, but somehow exerted control over cell wall loosening via some other mechanism. If growth can still occur under more alkaline conditions when ABA is not present, perhaps acidic pH causes an increase in ABA, which then leads to decreased cell wall loosening, perhaps through ABA regulation of the activity of expansins or xyloglucans. Another hypothesis is that perhaps ABA directly inhibits the uptake of sucrose and potassium via uptake co-transport mechanisms, rather than inhibiting proton pumps, affecting osmolarity and turgor.

Sensitivity of flowers, immature fruits and seeds to drought also plays a vital role in determining yield. The health and development of these tissues are essential for ensuring yield production, since they are what give rise to mature fruits and seeds. Yet, these tissues are highly susceptible to abortion when exposed to water deficits. As with other sinks, development of flowers and young fruits/seeds is slowed or halted under drought even when resources are not limiting. In common bean, this has been largely attributed to decreased cytokinin levels in flower and developing pods [70] which we know can be down-regulated by ABA. Abortion of pods is much less likely to occur after seed cell expansion and filling phase begin, because seed filling is not dependent on cytokinin activity. However, even though abortion of flowers and pods can be strongly increased under drought, high remobilization efficiency remains the strongest predictor of yield under drought in common bean, more so than pod or seed number. This may be due to two factors. First, common bean consistently produces many more flowers than will ever become mature fruits with seeds [71]. In fact, seed yield was only slightly lowered, or even increased, in common bean plants which had all their flowers removed for the first 15 days following anthesis. Therefore, there is a large buffer, such that many flowers can abort without affecting yield. This is true in both well-watered and water-limited conditions [72]. Therefore, even when abortion does occur, yields are not impacted as long as the number aborted does not go above a certain point. Second, imperfect synchronization of flowering, fruit and seed development increases the chance that some flowers and pods will develop under more favorable conditions. Together, these make it so pod number, seed number, and flower abortion rates are not very strong indicators of yield under stress. Instead, PHI correlations suggest that most plants end up with sufficient pods and seeds to fill and that the major limit on bean yield is filling of the seeds.

## 5. Drought Tolerance

While drought can negatively impact yield via carbon starvation [73]. the ability to maintain higher photosynthetic rates under drought alone does not impart drought tolerance. Instead, there exists a finer scale ability for plant to utilize available resources which contributes to increased drought tolerance. This is either through use of storage reserves or breaking down of structures in existing tissues, which can be remobilized to fruits and seeds. Plants which are better able to remobilize what is available to them under water deficits do so by maintaining higher sink strength, creating a stronger pull on those resources. Additionally, lines which maintain a higher sink strength are the lines which achieve higher yield under drought and are therefore deemed drought tolerant. In many agricultural plants, markedly improved yields have been associated, not with an increase in total biomass production, but a greater partitioning of the available carbon to the organs being harvested [74,75]. Additionally, in common bean specifically, the metric which shows the strongest correlation to yield under drought is pod harvest index (PHI), which quantifies remobilization efficiency of total pod resources into seeds [11,76].

Yet how inherent sink strength capacities are differently established and regulated between species, genotypes, or even different tissues within the same plant, is not well understood. While studies show differential growth and seed filling rates are determined by cell expansion and resource uptake rates, little is known about how the regulation of these rates differ between genotypes. However, ABA can be linked to these processes through its inhibition of mechanisms that promote expansion and resource uptake, and ABA typically increases under drought. This presents a possible source of regulation, where differences between genotypes could be due to differential sensitivities of ABA response or ABA production itself.

Plants evolved to sense and respond to their environment as a way to optimize their fitness. In the case of drought, this is usually achieved through ABA action. Although ABA has historically been considered a drought tolerance factor, Blum [24]. points out that whether ABA helps or hinders a plant depends on many things. For common bean, wild plants are indeterminate vines living for 8–10 months. They germinate and grow rapidly during the rainy season, developing leaves, flowers, pods and seeds until they sense the dry season beginning. When this occurs, the plants abort flowers, reduce vegetative growth and seed filling, and wait until the rainy season returns (Figure 2). Once rains return, the plants re-initiate both vegetative and reproductive growth, finishing their life cycle by producing beans. However, this response of pausing development during drought has unfortunately been carried into domesticated breeding lines, with the result that mild drought causes plants to slow leaf growth, abort flowers, and slow filling of existing pods and seeds. Therefore, ABA acts as a hindrance to yield in domesticated bush beans which cannot re-initiate reproductive growth like wild lines do. This can cause large impacts on yield, translating to decreased food security for those who depend on beans for protein and sustenance.

However, breeders have produced many lines that are able to perform well under drought. Unlike wild lines, these lines maintain higher rates of leaf growth, pod growth, and seed filling under drought. This results in higher seed yields in these lines. Perhaps in bean lines that are able to maintain higher leaf growth and seed filling under water deficits, their ability to sense or respond to ABA has been lost. It is possible some genotypes produce less or no ABA under drought, rendering them largely drought-insensitive. This would result in many downstream signaling responses not triggering under drought, so less changes in metabolism and uptake would occur. Alternatively, any of those individual responses downstream the ABA signaling pathway could become insensitive to ABA, even when ABA is present. This could lead to partial ABA insensitivity, reducing some of the ABA-induced drought response mechanisms. If common bean lines which maintain higher PHI under drought have become in some way insensitive to ABA signaling, this could explain why seed filling and leaf growth rates are higher under drought in tolerant lines than sensitive lines. If ABA is not present or sensed such that it inhibits SUT transport or proton pumping, filling rates could remain higher. However, if true that drought tolerant bean has become less sensitive to ABA or its downstream affects, this might also mean that stomata do not close and tolerant lines have increased water loss. Loss of ABA sensitivity in pepper resulted in increased water loss which lead to increased sensitive to drought [77]. Additionally, as mentioned above ABA is often seen as a drought tolerance factor, and increasing ABA sensitivity in some species, such as *Arabidopsis thaliana*, actually increases drought tolerance [78]. This opens many questions as to how ABA could result in opposite responses between different species, increasing drought tolerance in some and sensitivity in others.

With studies showing that leaf size, seed size, sucrose uptake rates, and other sink strength indicators all correlate with yield, it is clear that establishing and maintaining strong sink strength is a large component of achieving higher yields in common bean. Additionally, indeed, bean breeders are strongly focused on finding lines that have higher harvest indexes, namely PHI. However, much work is still needed to understand control of sink strength, the mechanisms drought impacts and how, and whether, as proposed, loss of ABA signaling and/or sensitivity acts as a factor in maintaining sink strength and therefore increasing drought tolerance in domesticated common bean. Better understanding of this control will hopefully help improve breeding efficiency in common bean as well as contribute to a better scientific basis for the understanding of sink strength in crops.

## Figures and Tables

**Figure 1 plants-10-00489-f001:**
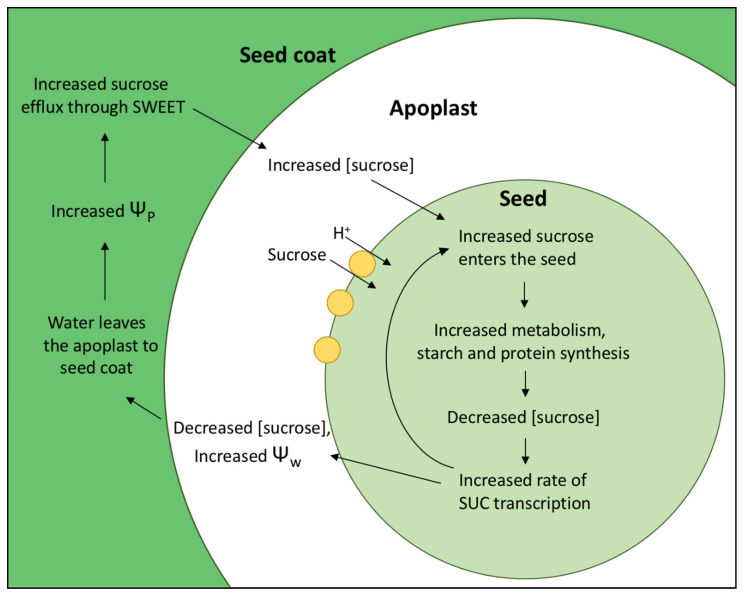
Regulation of sucrose uptake rates via sugar sensing feedback. SUC transcription is upregulated when low sucrose concentrations are sensed within the seed. This leads to increased levels of resources within the seed for metabolism and synthesis. When sucrose uptake rates increase, this leads to decreased levels of sucrose in the apoplast, increasing solute potential and therefore water potential (Ψ_w_). Water flows into the seed coat, down the water potential gradient. The resulting increase in turgor (Ψ_p_) in the seed coat increases sucrose efflux into the apoplast via channels called SWEETS. Increased sucrose in the apoplast leads to increased uptake via SUC, starting the cycle again.

**Figure 2 plants-10-00489-f002:**
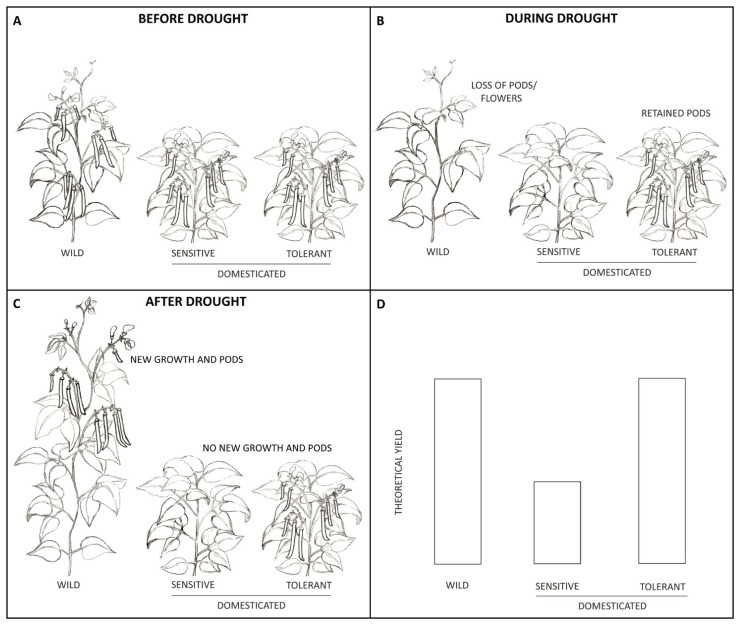
Effects of drought on flowers, pods, and yield in wild and domesticated common bean. (**A**) shows reproductive development in wild, vine beans compared to domesticated bush beans. Both have flowered and are producing pods. In (**B**), drought has caused both the wild and drought-sensitive domesticated lines to abort flowers and pods, however, the drought-tolerant line has retained its pods. All flowers and pods have been removed from these plants to accentuate drought’s affect, however, many times not all flowers and pods aborted under drought. (**C**) shows the wild line re-initiating both vegetative and reproductive growth once the drought has passed, allowing it to set new flowers and pods while the domesticated drought-sensitive line is unable to do the same. Theoretical differences in yields under drought (**D**) show that both the wild and drought-tolerant line are able to achieve a higher yield than the drought-sensitive line.

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
