# Peer review of "Sink Strength Maintenance Underlies Drought Tolerance in Common Bean"

_plants, 2021, doi:10.3390/plants10030489_

Round 1
Reviewer 1 Report
This manuscript aims at evaluating the implication of sink strength in maintaining yields of common bean under drought. Overall, authors have deeply analyzed this highlighting some interesting aspects such as i) the limited role of source function on yield maintenance under drought, ii) wild ancestors’ traits´ influence on drought stress responses in domesticated lines and iii) the key role of sink strengthening when aiming at improving yield performance under this abiotic stress.
My comments after revising this manuscript are:
- When authors affirm “If growth can still occur under more alkaline conditions when ABA is not present, perhaps acidic pH causes an increase in ABA, which then leads to decreased cell wall loosening, perhaps through ABA regulation of expansins or xyloglucans.” it is unclear to me till which extend ABA affects xylogucans (synthesis? Degradation?)
- Line 397-398: authors relate the higher growth rates and seed filling with the possibility of lacking ABA sensing. Have the authors checked if there is any data available in these bred lines that support this hypothesis?
- Figures 1 and 2 should be improved. Figure 1 could be presented in a more elaborated way (as it is more a summary scheme that a model). Figure 2 is a bit imprecise (labels indicating the sensitive, tolerant and wild cultivars should be indicated in all panels. Panel D is not adding any valuable information as it is presented in panels A to C. It would be interesting to include plant, pod and yield biomass in this figure. References used to elaborate this Figure should be included in the Figure legend).
Author Response
With respect to Reviewer 1:
We mention a possible explanation for a contradiction in the literature regarding ABA action, but we do not know of any data relevant to these ideas in bean or similar crops.
We edited Figure 2 as requested, adding wild, sensitive and tolerant to each panel. We did not remove panel D as the figure became lopsided, and we value the presentation in that panel. Biomass data for wild bean plants are not available to us.
Reviewer 2 Report
Manuscript of Hageman and Volkenburgh is a review on the role of sink strength maintenance in drought tolerance. In the manuscript the data on the mechanisms underlying sink reaction on drought stress in common bean are summarized. Given that the different lines of common beans show very different sink strengths under drought, which directly affect yield, it definitely makes actual to form a complete picture in a review article.
However, I have some minor remarks:
*Page 4, line 177: not “H+ ATPases”, but “H+-ATPases”
*Pages 7-8, lines 286, 300, 368 are empty
*Page 8, line 353: “(eg. Zinselmeier, Lauer, and Boyer 1995)” better is “(eg. Zinselmeier et al. 1995)”
*Page 6, lines 250-254: add “ΨP”, “ΨW” to figure legend and explain “SWEET”
*Page 6, line 252: “When sucrose uptake rates increase, this leads to decreased levels of sucrose in the apoplast, decreasing solute potential and therefore water potential.” But (page 5, line 237) “which lowers the sucrose concentration in the apoplast and increases water potential”.
Author Response
We have made minor revisions as specified by Reviewer 2; these are visible by viewing the attached manuscript in Track Changes.
*Page 4, line 177: not “H+ ATPases”, but “H+-ATPases” -- change made as requested
*Pages 7-8, lines 286, 300, 368 are empty – lines removed
*Page 8, line 353: “(eg. Zinselmeier, Lauer, and Boyer 1995)” better is “(eg. Zinselmeier et al. 1995)” – change made as requested
*Page 6, lines 250-254: add “ΨP”, “ΨW” to figure legend and explain “SWEET” – change made as requested
*Page 6, line 252: “When sucrose uptake rates increase, this leads to decreased levels of sucrose in the apoplast, decreasing solute potential and therefore water potential.” But (page 5, line 237) “which lowers the sucrose concentration in the apoplast and increases water potential”. – we corrected the legend by changing the word ‘decreasing’ to ‘increasing’ which makes all the rest consistent.
Reviewer 3 Report
This is very interesting work. My opinion is that you could be addressed the physiological, photosynthesis and intercellular, aspects of sink strength with structure or diagram on the manuscript and; if it does, the manuscript could greatly increase the understanding of the presentation of your work to the scientific community.
Please check the abstract and reference according to the journal.
Author Response
For Reviewer 3 we have not added another figure, as we feel Figure 1 addresses the point made.